# SEMANTIC-GUIDED LoRA PARAMETERS GENERATION

## ABSTRACT

Generating new Low-Rank Adaptation (LoRA) weights from pre-trained LoRAs has demonstrated strong generalization capabilities across a variety of tasks for efficiently transferring AI models, especially on resource-constrained edges. However, previous studies either merge base LoRAs via weighting coefficients or train a generative model in the closed-world assumption, limiting their efficiency and flexibility in complex edge user cases. This challenge may further increase when there are significant domain shifts between training and deployment. To this end, we propose **Semantic-guided LoRA Parameter Generation (`SG-LoRA`)**, a tuning-free generative framework to efficiently produce task-specific parameters for unseen tasks in a semantic-to-LoRA pipeline. Concretely, `SG-LoRA` uses task descriptions as the semantic bridge, measuring their proximity to a set of known expert tasks in a shared embedding space. Based on this semantic guidance, it models the target task's LoRA parameter distribution to generate high-performing parameters for novel tasks. `SG-LoRA` enables the real-time construction of LoRA models aligned with individual intents by distilling knowledge from prominent LoRA experts and, meanwhile, offering a privacy-preserving solution for personalized model adaptation in a novel zero-shot open-world setting proposed in this work. Extensive experiments on multiple challenging tasks confirm the superior performance and remarkable adaptability of `SG-LoRA`.

## 1 INTRODUCTION

In recent years, deep learning has seen remarkable progress, largely driven by the advent of large-scale pre-trained models (LPMs) Chung et al. (2024); Li et al. (2022); Liu et al. (2023); Rombach et al. (2022); Touvron et al. (2023). Trained on massive and diverse datasets, these models demonstrate exceptional performance across a wide range of downstream tasks Shenaj et al. (2024); Alayrac et al. (2022); Touvron et al. (2023). However, as both model and data scales continue to grow, retraining the entire model becomes increasingly computationally expensive and often infeasible in practice. To mitigate this challenge, parameter-efficient fine-tuning (PEFT) methods have drawn considerable attention Zhang et al. (2023c;b); Ding et al. (2023). Among them, Low-Rank Adaptation (LoRA) Hu et al. (2022) has emerged as a prominent approach. LoRA adapts pre-trained models by introducing a small number of trainable low-rank matrices into existing layers, achieving strong task-specific performance while leaving the original model weights unchanged Sung et al. (2022).

While an increasing number of pre-trained LoRA modules are becoming publicly available, effectively leveraging them in real-world scenarios remains a significant challenge. As shown in Figure. 1(a), we propose the Zero-Shot Open-world Adaption (ZSOA) in this paper, which aims to generate LoRA weights for unseen tasks based on a set of pre-trained LoRAs. ZSOA emphasizes two key aspects: *(1) No raw data is available for the unseen task, highlighting the need for rapid adaptability to evolving user intents* ; and *(2) Open-world task coverage, defined by a broad and unconstrained task space in which the unseen tasks may not be directly related to the seen tasks.* Compared to traditional LoRAs that need to be fine-tuned on downstream tasks, ZSOA shows data and computation-friendly strength, resulting in more flexibility in practice, particularly in edge environments where data privacy constraints and limited computational resources make large-scale retraining infeasible.

To enable broader applications of LoRA, prior research has explored two main directions, as illustrated in Figure. 1 (b-c), each partially addressing the challenges of ZSOA. The first line of work focuses on merging-based methods, which aim to rapidly construct task-specific models by directly fusing existing LoRA modules at hand Wortsman et al. (2022a); Yadav et al. (2023); Shenaj et al.

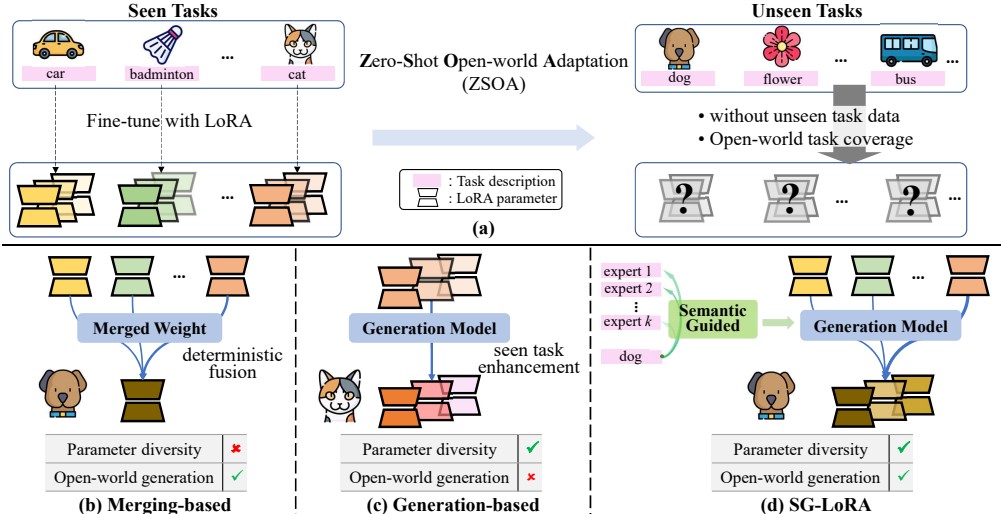

Figure 1: **Motivation of our SG-LoRA.** We consider a challenging scenario termed Zero-Shot Open-World Adaptation (ZSOA), where a model is provided rich LoRA resources for seen tasks but lacks access to data for unseen tasks during inference, with an unconstrained task space. Conventional LoRA adaptation methods are not suitable for ZSOA: merging-based approaches struggle to explore the diversity of LoRA parameters, while generation-based methods primarily focus on LoRA enhancement for seen task. Our SG-LoRA uses task descriptions as semantic guidance to enable conditional LoRA generation for unseen tasks in a data-free and open-world manner. (Each color family represents a set of LoRA parameters for the same task, for example, brown represents LoRAs for the *'Dog'* task, and yellow represents LoRAs for the *'Car'* task.)

(2024). Although these methods support open-world generation, the generated weights are obtained through deterministic fusion of existing LoRAs, resulting in limited diversity and constraining the model's ability to adapt to flexible or evolving requirements Shenaj et al. (2024); Wu et al. (2024). Moreover, the merging process must be carefully designed, as conflicts may arise when integrating LoRA modules trained on different tasks Zou et al. (2025); Zhao et al. (2024). In parallel, another direction explores generation-based methods, which leverage generative models, such as variational autoencoders (VAEs) Kingma & Welling (2013) or diffusion models Ho et al. (2020), to synthesize new LoRA parameters. By introducing stochasticity, these approaches enable greater diversity in parameter generation and bypass traditional fine-tuning pipelines. However, their success often relies on a closed-world assumption, where training and test tasks are drawn from similar distributions Soro et al. (2024). As a result, these methods are susceptible to task (or domain) shifts scenarios, and struggle to handle open-world tasks.

In this work, we draw inspiration from the human ability to intuitively infer semantic relationships between prior knowledge and new tasks Lake et al. (2017), enabling effective generalization in unfamiliar situations. For example, after learning to recognize cat breeds such as *Birman* and *Egyptian Mau*, a person can identify *British Shorthair* based solely on its textual description by relating it to previously acquired concepts. Motivated by this analogy-driven reasoning process, we propose a **S**emantic-**G**uided **LoRA** Parameter generation framework (SG-LoRA) that adapt LPMs to the ZSOA setting. Specifically, given a set of expert (or base) LoRAs whose weights are trained on seen tasks, we aim to train a semantic-to-LoRA model that takes the semantics of the unseen tasks as inputs and outputs high-performing LoRA weights directly. Notably, the task semantics serve as a bridge between the seen and unseen tasks, guiding our SG-LoRA on how to leverage expert knowledge to generate task-specific LoRA weights. Motivated by prompt engineering Zhou et al. (2022), we adopt the task description to identify the task semantics. These descriptions, typically concise yet semantically rich, are processed by a frozen CLIP text encoder to capture task-level correlations without exposing user-specific data. The task semantics are then modeled as Gaussian distributions according to the relationship between the unseen and seen tasks.

Importantly, simply scaling the number of expert LoRAs does not guarantee performance gains, as they may provide contradictory or irrelevant task knowledge. To resolve this, we design a sparse aggregator that assembles the most semantically relevant expert to integrate rational prior knowledge for a target task. During inference, the system directly generates target LoRA modules aligned with user requirements using only textual task queries. Crucially, the stochastic nature of our trained

generator converts deterministic LoRA construction into probabilistic parameter sampling, enhancing both parameter diversity and dynamic adaptability to evolving user intents. In summary, the main contributions of this work include :

- We introduce Semantic-Guided LoRA Generation, a versatile framework that harnesses semantic task relationships to enable zero-shot open-world adaptation. By conditioning on prior available task knowledge, `SG-LoRA` can synthesize high-performance LoRA parameters for arbitrary unseen tasks without retraining.

- By seamlessly integrating generated LoRA modules into off-the-shelf LPMs, our method enables fast personalization at inference time. It allows flexible configuration of the expert LoRA repository, supporting task-adaptive LoRAs both within and across datasets, thereby achieving scalable and adaptable model behavior.

- Comprehensive experiments on multiple image-text retrieval benchmarks demonstrate that the proposed method can rapidly generate LoRA parameters achieving performance comparable to traditional LoRA fine-tuning.

## 2 RELATED WORK

### 2.1 LOW-RANK ADAPTATION

Low-Rank Adaptation (LoRA) is a parameter-efficient fine-tuning method to adapt large models to novel tasks by approximating weight updates with low-rank matrices, drastically reducing trainable parameters while maintaining pre-trained knowledge Huang et al. (2023); Zhang et al. (2023a). Specifically, given a pre-trained weight $W_0$ with input $x$ and hidden state $h$, LoRA decomposes the weight update $\Delta W \in \mathbb{R}^{a \times b}$ into two low-dimensional matrices $B \in \mathbb{R}^{a \times r}$ and $A \in \mathbb{R}^{r \times b}$:

$$h = W_0 x + \Delta W x = W_0 x + \gamma B A x, \tag{1}$$

where the rank $r \ll \min(a, b)$. $\gamma$ is a constant scaling hyperparameter that controls the contribution of the LoRA update. Recently, a growing number of high-quality, pre-trained LoRA modules have become publicly available for various architectures Civitai (2024); HuggingFace (2024), including transformers and vision–language models, offering rich resources to accelerate adaptation and deployment on specific downstream tasks.

### 2.2 MODEL MERGING

Model merging integrates parameter-level knowledge from multiple independently trained networks into a unified model, achieving enhanced capabilities Li et al. (2023); Ilharco et al. (2022); Jang et al. (2024); Wortsman et al. (2022b). The pioneer work Model Soups Wortsman et al. (2022a) establishes weight averaging as a foundational paradigm, showing that averaging fine-tuned models from identical pre-trained bases with varied hyperparameters consistently outperforms individual models. AdapterSoup Chronopoulou et al. (2023) generalizes the Model Soups paradigm to cross-domain adaptation by dynamically averaging domain-specific adapters at test time. This approach preserves the base model's integrity while enhancing out-of-distribution generalization through selective weight-space interpolation of relevant domain knowledge. Recent advances have extended this paradigm to LoRA-based module fusion. For instance, LoraHub Huang et al. (2023) dynamically composes pre-trained LoRA modules by optimizing their weights through few-shot examples from new tasks, leveraging black-box optimization techniques (e.g., CMA-ES) to achieve efficient adaptation without backpropagation. Meanwhile, SemLA Qorbani et al. (2025) introduces a training-free approach by directly comparing test images' visual features with known domain prototypes, using the resulting similarity to efficiently guide adapter retrieval and fusion. However, they either require unknown-task data or involve loading and unloading multiple LoRA adapters for each input, which can be computationally impractical. Moreover, they rely on inflexible, deterministic fusion for unseen tasks.

### 2.3 NEURAL NETWORK PARAMETERS GENERATION

Although generative modeling has advanced considerably, the direct generation of network weights for pre-trained models remains an emerging area of study Knyazev et al. (2021); Zhmoginov et al. (2022);

Knyazev et al. (2023); Wang et al. (2024a). Approaches such as generative hyper-representation learning Ha et al. (2016), neural network diffusion Peebles et al. (2022); Hu et al. (2021); Jin et al. (2024), and kernel density estimation–based methods Soro et al. (2024) have shown promise but remain fundamentally limited to small architectures and unconditional weight generation within fixed distributions. Consequently, these methods struggle to generalize to unseen tasks, constraining their broader applicability. While meta-learning frameworks Nava et al. (2022); Zhang et al. (2024) have enabled powerful joint model generation for visual recognition and few-shot learning, they often neglect the personalization and parameter diversity, restricting the generator's output to classifier heads rather than more flexible and expressive parameter sets, such as LoRA. ICM-LoRAShao et al. (2025) innovatively explores the parameter relationships among tasks through task vectors, but focuses on closed-world, task-specific enhancements of LoRA parameters. As a result, the question of whether one can rapidly generate efficient, user-intent–focused LoRA parameters in open-world settings remains unexplored.

## 3 THE PROPOSED MODEL

In this section, we begin with an overview of essential concepts for understanding semantic-guided LoRA parameter generation, followed by detailed descriptions of our proposed methods.

### 3.1 PROBLEM DEFINATION AND PRELIMINARY

We define Zero-Shot Open-world Adaptation (ZSOA) as a novel and challenging task setting that requires models to generalize across semantically diverse tasks in open-world scenarios. Unlike conventional zero-shot learning or task transfer—often confined to fixed label spaces or narrow domains—ZSOA focuses on tasks that share a common structural format (e.g., image-text retrieval) but differ substantially in domain, content, or distribution. It emphasizes that queried tasks during inference time are drawn from an undefined and unbounded set, requiring rapid adaptation to novel tasks without access to raw data. This setting reflects realistic deployment scenarios, where task-level generalization must rely solely on prior experience.

In this work, we instantiate ZSOA in the context of fine-grained image-text retrieval, with each task formulated as a retrieval problem over a specific semantic category (e.g., animal species, flower type). Formally, given a set of fine-tuned LoRA module $\mathcal{W}$ trained on known tasks $\mathcal{T}$, our goal is to adapt the model to unseen task $\mathcal{T}^*$ without accessing any labeled image-text pairs. Let $f(\mathcal{T})$ denote the textual description of task $\mathcal{T}$, we learn a generator $G$ that predicts LoRA parameters for $\mathcal{T}^*$ based on semantic descriptions and $\mathcal{W}$:

$$\mathcal{W}^* = G\left(f(\mathcal{T}^*), \mathcal{W}, f(\mathcal{T})\right) \tag{2}$$

The synthesized LoRA parameter $\mathcal{W}^*$ is then used to modulate a frozen vision-language backbone, enabling it to perform the image-text retrieval task defined by $\mathcal{T}^*$. ZSOA thus extends traditional zero-shot learning by enabling parameter-level generalization , accommodating diverse user-instructed tasks with open-world queries.

### 3.2 LORA PARAMETER DATASET CONSTRUCTION

#### 3.2.1 TASK-SPECIFIC LORA TRAINING

The first stage involves constructing a dataset of LoRA parameters. Consider a collection of $N$ distinct tasks $\mathcal{T} = \{\mathcal{T}_1, \ldots, \mathcal{T}_N\}$, where each $\mathcal{T}_n$ corresponds to a specific retrieval task (e.g., image-text retrieval for *Cat* category). In this context, the task is dataset-agnostic—that is, it may originate from the same dataset or from a different one. Given a pre-trained vision-language model (VLM), we train a task-specific LoRA for each $\mathcal{T}_n$ using corresponding image-text pairs, applying LoRA modules at consistent positions within the VLM. To ensure comparability and reduce variance, all LoRA modules are trained using identical network configurations by default. After training stabilizes, we extract and store the LoRA from the final $M$ epochs, yielding task-specific parameter data:

$$\Delta\mathbf{W}_n = \{\Delta\mathbf{W}_n^m\}_{m=1}^M, \quad \mathbf{d}_n = f(\mathcal{T}_n), \tag{3}$$

where $\Delta\mathbf{W}_n^m = [\boldsymbol{B}_n^m, \boldsymbol{A}_n^m]$ denotes the concatenation of LoRA parameters in layer-wise order for task $n$ from the $m$-th saved epoch. $\mathbf{d}_n$ denotes the textual description associated with $\mathcal{T}_n$, generated

using the template *'a photo of a <class name>'*, and encoded by the frozen CLIP text encoder $f(\cdot)$ to serve as a global semantic representation. Collectively, these form the LoRA parameter dataset:

$$\mathcal{W} = \{\Delta\mathbf{W}_1, \ldots, \Delta\mathbf{W}_N\}, \tag{4}$$

where each element is optimized for image-text alignment within its respective task.

### 3.2.2 LoRA EXPERT REPOSITORY FORMATION

To construct a reliable expert LoRA space that simulates diverse LoRA resources, we first curate a representative subset of tasks from the full corpus $\mathcal{W}$, where each task is associated with its corresponding LoRA parameters and semantic embedding. For each selected task, we compute the mean LoRA parameters $\boldsymbol{\mu}_e$ over all available adaptations, yielding a distilled representation of its task-specific adaptation pattern. These averaged parameters, along with their associated semantic embeddings, constitute our expert repository:

$$\mathcal{W}_{\text{expert}} = \{(\boldsymbol{\mu}_e, \mathbf{d}_e) \mid e \in \mathcal{E}\}, \quad \boldsymbol{\mu}_e = \frac{1}{M}\Delta\mathbf{W}_e, \tag{5}$$

where $\mathcal{E}$ represents the selected expert index and $\mathcal{W}_{\text{expert}}$ serves as a compact yet expressive basis for capturing the essential characteristics of each knowledge domain. The remaining LoRA data are then split into training and evaluation sets for subsequent model development and evaluation.

### 3.3 SEMANTIC-GUIDED LoRA PARAMETER GENERATION

Given the collected LoRA repository $\mathcal{W}_{\text{expert}}$, our SG-LoRA framework generates conditional LoRA parameters under semantic guidance. We first define the task semantics by selecting and combining the most relevant expert LoRAs from the repository using a sparse aggregator. A conditional variational autoencoder (CVAE) Sohn et al. (2015) is then trained to generate target LoRA parameters aligned with the corresponding task semantics. Once trained, the model can generalize to unseen tasks in an open-world query setting without further training.

#### 3.3.1 CONSTRUCTION OF TASK SEMANTICS

Intuitively, not all experts contribute equally to an unseen task. Therefore, it is essential to identify and prioritize the most beneficial experts. Fortunately, the CLIP textual encoder is well-suited for this purpose, as it effectively captures rich semantic relationships across tasks. As shown in Eq. 3, we use each task's textual embedding as a global semantic descriptor for its LoRA parameters. For an unseen task $\mathcal{T}^*$ with textual embedding $\mathbf{d}^*$, we compute cosine similarities with all expert embeddings $\{d_e | e \in \mathcal{E}\}$ and select the top-$K$ experts with the highest similarity scores to form a semantically tailored expert set, indexed by $\mathcal{I}_{\text{top-}k}$. Then, we normalize their similarity scores using the softmax function to obtain the fusion coefficients:

$$\alpha_k = \frac{\exp\left(\text{sim}(\mathbf{d}^*, \mathbf{d}_k)/\tau\right)}{\sum_{k' \in \mathcal{I}_{\text{top-}k}} \exp\left(\text{sim}(\mathbf{d}^*, \mathbf{d}_{k'})/\tau\right)}, \quad k \in \mathcal{I}_{\text{top-}K}, \tag{6}$$

where $\tau > 0$ is a temperature parameter. The semantic vector for task $\mathcal{T}^*$ is computed as a weighted sum:

$$\boldsymbol{\mu}^* = \sum_{k \in \mathcal{I}_{\text{top-}K}} \alpha_k \cdot \boldsymbol{\mu}_k. \tag{7}$$

The attention strategy in Eq. 7 guides our model to understand the unseen task with the expert LoRAs from the textual perspective, resulting in high-quality semantic representation.

To capture the semantic diversity of the task, we also consider estimating the element-wise variance for task $\mathcal{T}^*$ under the Law of Total Variance theory:

$$\boldsymbol{\sigma}^{*2} = \sum_{k=1}^{K} \alpha_k \boldsymbol{\sigma}_k^2 + \sum_{k=1}^{K} \alpha_k (\boldsymbol{\mu}_k - \boldsymbol{\mu}^*) \odot (\boldsymbol{\mu}_k - \boldsymbol{\mu}^*), \tag{8}$$

where $\odot$ denotes element-wise multiplication and $\boldsymbol{\sigma}_k$ denotes the variance of the $k$-th expert. This flexible formulation enables the model to better reflect the statistical properties of new tasks, which is crucial for generative modeling. The estimated mean and variance guide the generative process, resulting in more accurate and context-aware outputs, thereby improving generalization and robustness. We leave the derivation of Eq. 8 in Appendix A.2. To simplify, we will use $c$ to represent the task semantics of $\mathcal{T}^*$ in the following descriptions, *e.g.*, $c = \{\boldsymbol{\mu}^*, \boldsymbol{\sigma}^{*2}\}$.

### 3.3.2 CONDITIONAL LoRA PARAMETER GENERATION

We adopt a conditional variational autoencoder framework to generate target LoRA parameters based on the task semantics calculated above. Given a batch of training LoRA tensor $\boldsymbol{X}$, the encoder approximates the posterior distribution $q(\boldsymbol{z}|\boldsymbol{X}, c)$ using a multi-layer perceptron (MLP) that takes the $\boldsymbol{X}$ to be reconstructed and the task semantics $c$ as input. A latent code $z \sim q(\boldsymbol{z}|\boldsymbol{X}, c)$ is sampled and passed to the decoder along with $c$ as condition to reconstruct the original input $\boldsymbol{X}$. Unlike traditional VAEs that adopt $p(\boldsymbol{z}) = \mathcal{N}(0, \mathbf{I})$ as the prior distribution, we here develop a semantic-aware prior for each task $p(\boldsymbol{z}|c)$. $p(\boldsymbol{z}|c)$ is parameterized with stacked MLPs, allowing the model to flexibly represent a task-specific prior distribution based on domain-level statistics.

The model is trained to maximize the evidence lower bound (ELBO), which consists of two terms: the reconstruction and the regularization term :

$$\mathcal{L}_{\text{CVAE}} = \mathbb{E}_{q(\boldsymbol{z}|\boldsymbol{X}, c)} \left[ \|\boldsymbol{X} - \hat{\boldsymbol{X}}\|^2 \right] + \lambda \cdot KL(q(\boldsymbol{z}|\boldsymbol{X}, c) \| p(\boldsymbol{z}|c)), \quad (9)$$

where $\hat{\boldsymbol{X}}$ denotes the reconstructed LoRA parameters, $KL(\cdot \| \cdot)$ is the Kullback-Leibler divergence, and $\lambda$ controls the relative weight of the *KL* term. The first term encourages the decoder to reconstruct accurate LoRA parameters, while the second term regularizes the latent space to align the task-specific prior. During inference, a sample $z$ is drawn from the prior distribution $p(\boldsymbol{z}|c)$, and the decoder generates the corresponding custom LoRA parameter.

## 4 EXPERIMENTS

### 4.1 EXPERIMENTAL SETTINGS

**Datasets.** The proposed SG-LoRA is evaluated on three benchmark datasets. Specifically, we use the widely adopted MS-COCO dataset Lin et al. (2014), a standard benchmark for image-text retrieval, known for its diverse scenes and rich linguistic annotations. To evaluate the model's generalization ability, we further include the OxfordPets dataset Parkhi et al. (2012) and the Flowers102 dataset Nilsback & Zisserman (2008), both of which are originally designed for fine-grained image classification. Moreover, given the inherent ambiguity and limited informativeness of MS-COCO captions and the absence of captions in the other two datasets, we construct synthetic textual descriptions using Qwen2-VL Wang et al. (2024b). For MS-COCO, we use the original training split but regenerate captions for each image, effectively creating a new image–caption dataset, and then divide it into training, validation, and test sets. More details are provided in Appendix A.1.

**Metrics.** The evaluation metric used is Recall@K (R@K), which quantifies the proportion of correct matches appearing in the top-K retrieved candidates. We report R@1, R@5, and R@10 for both image-to-text and text-to-image retrieval scenarios.

**Implementation Details.** We adopt CLIP ViT-B/16 as our backbone, injecting rank-2 LoRA adapters into the $W_q$, $W_k$, and $W_v$ projection matrices of every Transformer block in the visual encoder. Training is carried out with the Adam optimizer. The CVAE's encoder and prior network each consist of two-layer MLPs with ReLU activations, whereas the decoder is realized as a three-layer MLP with ReLU activations. We set the default values of $M$, $K$, and $\lambda$ in the model to 100, 4, and 1, respectively. All experiments were performed on a single NVIDIA A6000 GPU.

### 4.2 COMPARATIVE METHODS

To evaluate the effectiveness of the proposed method, we compare it with the following methods: (1) **Zero-Shot CLIP**: The original CLIP model without any adaptation of LoRA modules. (2) **Model Soups**: Consistent with Wortsman et al. (2022a) all LoRA experts in $\mathcal{W}_{\text{expert}}$ are uniformly averaged without considering their relevance to the target task. (3) **AdapterSoup** : The top-$K$ experts from the expert repository are selected based on the semantic similarity vector and with equal weight, assigning each a coefficient of $1/K$. This can be seen as an variant of Chronopoulou et al. (2023) revised to our setting. (4) **Top-$K$ LoRA Weighted**: The top-$K$ experts are selected based on the semantic similarity vector, and their weights are computed by applying a softmax function over the similarity scores for adaptive merging. (5) **SG-LoRA**: Our proposed method, which generates

task-specific LoRAs based on semantic proximity. (6) **Oracle:** For each task, LoRA parameters are trained individually on the specific dataset where we evaluate.

### 4.3 MAIN RESULTS AND DISCUSSION

#### 4.3.1 IN-DATASET IMAGE-TEXT RETRIEVAL

We first conducted in-dataset evaluations on MS-COCO and OxfordPets dataset separately, with results shown in Table.1. Several key observations are as follows: 1). Compared to the Zero-Shot CLIP baseline and consistent with previous findings Qorbani et al. (2025), directly merging all experts in the expert repository leads to performance improvements. 2). Selecting semantically relevant experts, like those most related to the current query task from the repository, can further enhance performance. However, naively treating all selected experts equally may result in degraded performance. For instance, in Table 1, AdapterSoup underperforms compared to Top-$K$ LoRA Weighted. This may be because different experts contribute unevenly to the target task. Assigning equal weight ignores these differences and may amplify noise from less relevant experts. By contrast, incorporating semantic weighting coefficients allows the fusion process to account for varying degrees of relevance, leading to more effective integration of expert knowledge and improved retrieval performance. 3). While merging-based approaches still fall short of the performance achieved by directly fine-tuning LoRA on the unseen task, SG-LoRA fully recovers the performance of oracle adapters.

Notably, SG-LoRA even outperforms Oracle in certain cases—for example, R@1 for bidirectional retrieval on MS-COCO and R@1 for image-to-text retrieval on OxfordPets. This improvement gains from the efficient compression of expert LoRAs: our trained CVAE integrates the target LoRA using compact yet semantically rich task representations, enabling the generation of target-aligned LoRA parameters by modeling their distribution in the parameter space. Moreover, the Oracle LoRA fine-tuned on unseen tasks sometimes suffers from overfitting, especially when trained on a small set of image-caption pairs. Our SG-LoRA helps mitigate this performance drop, likely due to its ability to generalize without relying on target-task data.

Table 1: Model Performance Comparison on MS-COCO and OxfordPets Datasets. The best results are highlighted in bold, and the second-best results are underlined.

| Method | MS-COCO | | | | | | OxfordPets | | | | | |
| | I2T | | | T2I | | | I2T | | | T2I | | |
| | R@1 | R@5 | R@10 | R@1 | R@5 | R@10 | R@1 | R@5 | R@10 | R@1 | R@5 | R@10 |
|---|---|---|---|---|---|---|---|---|---|---|---|---|
| Zero-Shot CLIP | 66.43 | 84.31 | 89.14 | 41.66 | 64.63 | 73.01 | 40.45 | 66.27 | 77.53 | 26.03 | 50.66 | 62.98 |
| Oracle | 72.45 | 88.91 | 93.41 | 53.10 | 76.47 | 83.97 | 55.84 | 81.84 | 89.13 | 40.99 | 70.41 | 80.39 |
| Model Soups | 69.37 | 85.96 | 90.95 | 47.38 | 69.54 | 77.97 | 52.54 | 77.80 | 85.59 | 33.51 | 61.77 | 72.93 |
| AdapterSoup | 70.70 | 86.57 | 91.09 | 48.64 | 70.51 | 78.79 | 52.59 | 78.52 | 86.09 | 34.05 | 62.70 | 73.93 |
| Top-$K$ LoRA Weighted | _71.55_ | _87.54_ | _91.69_ | _49.85_ | _71.79_ | _79.66_ | _53.96_ | _79.41_ | _86.53_ | _35.42_ | _64.08_ | _74.99_ |
| SG-LoRA | **74.31** | **88.78** | **92.50** | **54.42** | **75.45** | **82.18** | **57.15** | **80.40** | **88.04** | **37.62** | **67.16** | **77.44** |

#### 4.3.2 CROSS-DATASET IMAGE-TEXT RETRIEVAL

Given the flexibility of SG-LoRA, we conducted a more challenging cross-dataset evaluation. As shown in Table.2 and 6, SG-LoRA consistently outperforms merging-based approaches in these settings. Interestingly, we also observed that models trained on MS-COCO were able to generate LoRA parameters that, in some cases, outperformed those trained directly on the OxfordPets (e.g., a relative improvement of 1.22% in T2I R@1). This may be attributed to the richer expert knowledge available in MS-COCO, which, due to its broader data diversity, enables more extensive exploration of the parameter space during generation, a capability not achievable when generating within the narrower scope of OxfordPets. Another possible reason is that the uniform LoRA training configuration used across datasets (as described in Section 3.2.1) may not be optimal for OxfordPets. Conversely, we find that when the generation model is trained on OxfordPets and applied to MS-COCO, its performance is generally worse than that of models trained on MS-COCO.

Table 2: Cross-Dataset Generalization Performance: Bidirectional Evaluation between MS-COCO and OxfordPets. The best results are highlighted in bold, and the second-best results are underlined.

| Method | MS-COCO → OxfordPets | | | | | | OxfordPets → MS-COCO | | | | | |
| | I2T | | | T2I | | | I2T | | | T2I | | |
| | R@1 | R@5 | R@10 | R@1 | R@5 | R@10 | R@1 | R@5 | R@10 | R@1 | R@5 | R@10 |
|---|---|---|---|---|---|---|---|---|---|---|---|---|
| Zero-Shot CLIP | 40.45 | 66.27 | 77.53 | 26.03 | 50.66 | 62.98 | 66.43 | 84.31 | 89.14 | 41.66 | 64.63 | 73.01 |
| Oracle | 55.84 | 81.84 | 89.13 | 40.99 | 70.41 | 80.39 | 72.45 | 88.91 | 93.41 | 53.10 | 76.47 | 83.97 |
| Model Soups | 44.67 | 70.91 | 80.78 | 30.45 | 56.77 | 68.52 | 68.58 | 85.67 | 90.62 | 44.09 | 66.55 | 75.08 |
| AdapterSoup | 45.96 | 71.83 | 81.42 | 30.88 | 57.32 | 69.08 | 68.74 | 85.83 | 90.63 | 44.19 | 66.58 | 75.31 |
| Top-$K$ LoRA Weighted | 48.13 | 73.43 | 82.73 | 33.34 | 59.53 | 70.89 | 68.75 | 85.77 | 90.67 | 44.60 | 66.76 | 75.25 |
| SG-LoRA | 55.41 | 80.73 | 87.33 | 38.84 | 66.77 | 76.69 | 70.81 | 86.83 | 91.41 | 46.50 | 68.73 | 77.19 |

## 4.4 GENERALIZATION IN STANDARD IMAGE-TEXT RETRIEVAL

Considering another complex scenario where the test task may contain image-text pairs from multiple categories, we evaluated retrieval performance on the Flickr30K test set Plummer et al. (2015). Since this dataset has no clear category distinction during retrieval, we randomly selected one caption per image, fed it into the CLIP textual encoder to obtain the textual embedding, and then calculated the mean value as the task description for retrieval. The experimental results are shown in Figure.2, where SG-LoRA outperforms Zero-Shot CLIP. Additionally, SG-LoRA trained on MS-COCO achieves better results than that trained on OxfordPets. This is because MS-COCO provides more comprehensive expert knowledge, covering a wider range of categories, while OxfordPets primarily focuses on fine-grained distinctions within just two broad categories—cats and dogs. This also indicates that when semantic guidance is more powerful and comprehensive, or more relevant to the downstream task, the generated LoRA parameters are also superior.

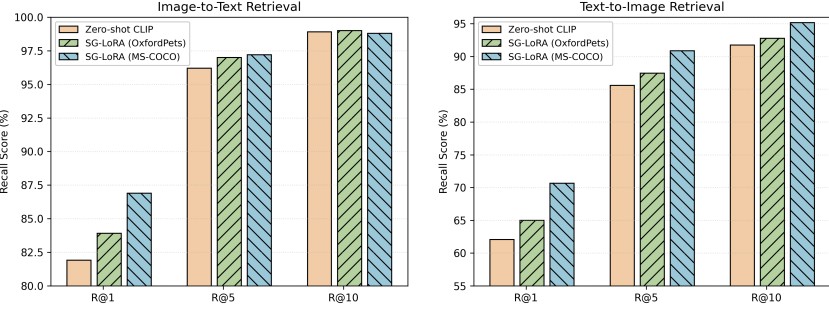

Figure 2: Simulation of task-agnostic evaluation on a general image-text retrieval dataset. We train models on MS-COCO and OxfordPets, respectively, and evaluate them on the Flickr30K test set.

## 4.5 ABLATION STDUY

**Impact of expert repository configuration.** In the experiment of Table. 2, we observed that the *Cat* expert from the MS-COCO dataset was frequently selected as a semantic condition. To further evaluate the impact of semantically salient experts, we assessed how the inclusion of the *Cat* expert from MS-COCO influences retrieval performance on two unseen cat classes on OxfordPets. As shown in Table. 3, incorporating the *Cat* expert into the expert repository improves performance in most cases, particularly for text-to-image retrieval. This highlights the effectiveness of semantically guided expert selection. This finding also demonstrates the flexibility of our method in constructing task-adaptive expert repositories—particularly when a richer pool of LoRA resources is available.

Table 3: Ablation on expert repository strategy for cross-dataset evaluation. We evaluate how the MS-COCO *Cat* expert affects retrieval on two unseen OxfordPets cat tasks (marked with gray text).

| Expert strategy | *Egyptian Mau* I2T | | | *Egyptian Mau* T2I | | | Expert strategy | *Persian* I2T | | | *Persian* T2I | | |
| | R@1 | R@5 | R@10 | R@1 | R@5 | R@10 | | R@1 | R@5 | R@10 | R@1 | R@5 | R@10 |
|---|---|---|---|---|---|---|---|---|---|---|---|---|---|
| w/o *Cat* expert | 36.08 | 62.89 | 71.13 | 15.21 | 31.70 | 44.07 | w/o *Cat* expert | 44.00 | 80.00 | 87.00 | 34.00 | 62.25 | 72.75 |
| w/ *Cat* expert | 37.11 | 63.92 | 72.16 | 15.21 | 35.05 | 46.91 | w/ *Cat* expert | 47.00 | 79.65 | 86.00 | 36.75 | 64.25 | 73.75 |

Table 4: Top-4 expert for *Yorkshire Terrier* category under mixed-source experts configurations

| Expert Task | Source Dataset | Contribution |
|---|---|---|
| *Scottish Terrier* | OxfordPets | 0.9221 |
| *Dog* | MS-COCO | 0.0692 |
| *Cat* | MS-COCO | 0.0082 |
| *American Bulldog* | OxfordPets | 0.0005 |

Given that our model supports open-world expert repository construction, we further conducted a case study where experts from OxfordPets, MS-COCO, and Flowers102 were combined into a mixed-source repository. We also combined training data from both datasets to train the `SG-LoRA` model accordingly. Figure.3 presents a comparison between single-source and mixed-source expert configurations. Additionally, we present the top-4 experts selected by `SG-LoRA` under the mixed-source setup for the unseen *Yorkshire Terrier* category, along with their corresponding weights in Table.4. As shown, the mixed-source experts yield better performance than the single-source experts. The performance even surpasses that of the oracle LoRA model in text-to-image retrieval. These demonstrate the potential of our method in more realistic, real-time application scenarios, where expert repositories are constructed dynamically from heterogeneous data sources.

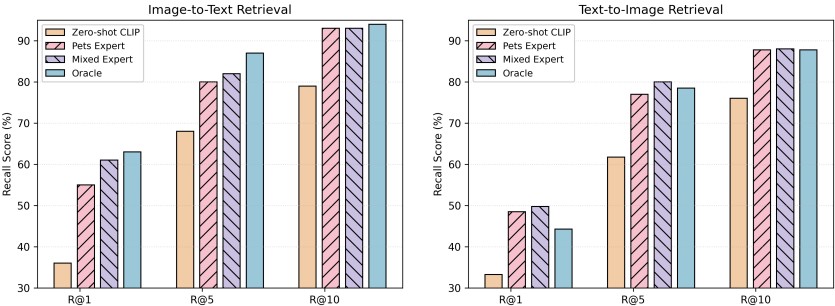

Figure 3: Comparison on expert repository configuration: single-source experts vs. mixed-source experts.

**Impact of the number of experts.** We conduct ablation studies on the number of experts $K$ used in task semantic construction. As shown in Figure.4, using too few experts leads to insufficient knowledge for generalizing to unseen tasks, while increasing $K$ incorporates more semantic information but may also introduce irrelevant context. Overall, $K = 4$ shows a good balance between expert diversity and relevance, delivering good performance on both datasets.

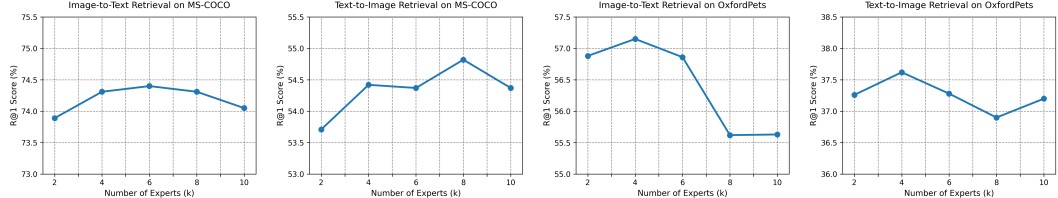

Figure 4: Ablaton study on the number of experts.

## 5 CONCLUSION

In this work, we introduce a novel and challenging task setting—Zero-Shot Open-world Adaptation (ZSOA)—which requires models to generalize across semantically diverse tasks in open-world scenarios. To achieve this, we propose a flexible and efficient approach that dynamically generates task-specific LoRA parameters guided by available LoRA resources. By identifying the most relevant expert knowledge based on semantic similarity and leveraging task semantics in a conditional generative framework, our `SG-LoRA` models the distribution of unseen parameters in a tuning-free manner. The inherent stochasticity of our generation process further introduces diversity, enhancing adaptability to previously unseen tasks. `SG-LoRA` is scalable and naturally privacy-preserving, making it well-suited for deployment in sensitive and dynamic real-world environments.

## REPRODUCIBILITY STATEMENT

We provide detailed descriptions of the model, training procedure, and evaluation in the main text. Additional implementation details, hyperparameters, and ablation studies are included in the Appendix A.

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

# A APPENDIX

## A.1 EXPERIMENTAL DETAILS

### A.1.1 IMAGE-TEXT RETRIEVAL DATASET

In this work, we generate fine-grained captions for the image-text retrieval task using Qwen2-VL. As illustrated in Figure.5, our in-context learning approach provides the Multimodal Large Language Model (MLLM) with a small number of demonstration examples, enabling it to generate detailed captions for target images. Using this method, we have produced four diverse and highly relevant captions for each image in the entire OxfordPets dataset and the Flowers102 dataset, a subset of the MS-COCO dataset. These fine-grained descriptions serve as a valuable resource for downstream tasks such as fine-grained image-text retrieval and facilitate further research in ZSOA.

---

**[Instruction]**
*You are an image description assistant. Please analyze the given image carefully and provide a detailed description of its contents.*

**[Demonstration 1]**
*Image: <image_path1>*
*Input: A photo of pug.*
*Output: The pug has a fawn coat, black mask, and wrinkled face with a short-muzzled expression. It wears a green collar with a tag, sits on a rock, and has a solemn gaze.*

**[Demonstration 2]**
*Image: <image_path2>*
*Input: A photo of birman.*
*Output: The Birman cat has a sleek cream and brown coat, striking blue eyes, and a dark brown nose. It wears a blue collar and is perched on a scratching post.*

*[Target Query]*
***Image: <image_path>***
*Input: A photo of {class name}.*
*Output: **fine-grained caption for the target image***

---

Figure 5: Examples of caption generation using Qwen2-VL.

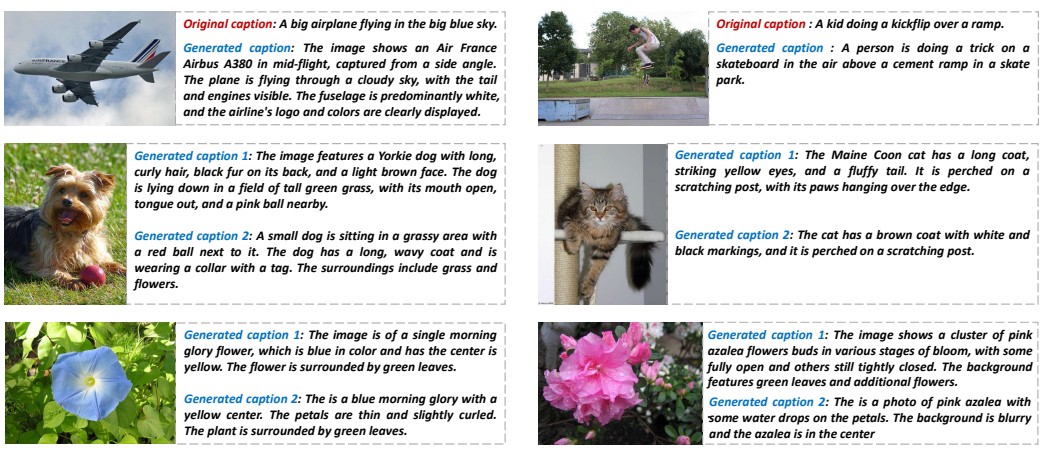

Figure 6: Illustration of generated captions: results on the MS-COCO (top), OxfordPets (middle), and Flowers102 (bottom) datasets.

As shown in Figure.6, we present examples of image-text pairs from three datasets. Compared to the original captions in the COCO dataset (labeled as *Original Caption* in the top), our generated captions more accurately reflect the content of the corresponding images. Thus, these datasets enable us to assess the model's robustness both in in-domain retrieval and in generalizing to novel domains and compositional scenarios.

Table 5: Selected Expert Tasks for Each Datasets

| MS-COCO | OxfordPets | Flowers102 |
|---|---|---|
| *Airplane* | *Abyssinian* | *Sweet pea* |
| *Truck* | *American bulldog* | *Tiger lily* |
| *Traffic Light* | *American Pit Bull Terrier* | *Monkshood* |
| *Cat* | *Birman* | *King Protea* |
| *Horse* | *Bombay* | *Corn Poppy* |
| *Giraffe* | *British Shorthair* | *Daffodil* |
| *Handbag* | *German Shorthaired* | *Sunflower* |
| *Snowboard* | *Havanese* | *Osteospermum* |
| *Wine Glass* | *Keeshond* | *Anthurium* |
| *Banana* | *Leonberger* | *Hibiscus* |
| *Hot Dog* | *Newfoundland* | *Desert-Rose* |
| *Laptop* | *Scottish Terrier* | *Mallow* |

### A.1.2 LORA EXPERT REPOSITORY DETAILS

We present the expert tasks included in each dataset's expert repository in Table. 5, as set in the main manuscript.

### A.2 AGGREGATION FUNCTION FOR SEMATIC PRIOR

We begin with the classical identity in probability theory known as the Law of Total Variance: for a random vector $\Theta$ and a discrete conditioning variable $C$, the following identity holds:

$$\text{Var}(\Theta) = \mathbb{E}_C\big[\text{Var}(\Theta \mid C)\big] + \text{Var}_C\big(\mathbb{E}[\Theta \mid C]\big), \tag{10}$$

**Proof Sketch:** Using the identity $\text{Var}(\Theta) = \mathbb{E}[\Theta^2] - (\mathbb{E}[\Theta])^2$ and the Law of Iterated Expectations, we write:

$$\mathbb{E}[\Theta^2] = \mathbb{E}_C\big[\mathbb{E}[\Theta^2 \mid C]\big], \quad \mathbb{E}[\Theta] = \mathbb{E}_C\big[\mathbb{E}[\Theta \mid C]\big], \tag{11}$$

From the definition of conditional variance, $\text{Var}(\Theta \mid C) = \mathbb{E}[\Theta^2 \mid C] - (\mathbb{E}[\Theta \mid C])^2$, we substitute and obtain:

$$\text{Var}(\Theta) = \mathbb{E}_C\big[\text{Var}(\Theta \mid C)\big] + \mathbb{E}_C[(\mathbb{E}[\Theta \mid C])^2] - (\mathbb{E}_C[\mathbb{E}[\Theta \mid C]])^2, \tag{12}$$

where the last two terms equal $\text{Var}_C(\mathbb{E}[\Theta \mid C])$.

**Task Semantic via Aggregation.** Suppose we have $K$ expert tasks, each providing a LoRA parameter set with mean $\boldsymbol{\mu}_i$ and variance $\boldsymbol{\sigma}_i^2$. For an unseen task $\mathcal{T}^*$ with expert weight vector $\boldsymbol{\alpha} = [\alpha_1, \ldots, \alpha_k]^T$, where $\alpha_i \geq 0$ and $\sum_i \alpha_i = 1$, we construct the prior as a weighted combination. The prior mean is:

$$\boldsymbol{\mu}^* = \sum_{i=1}^k \alpha_i \boldsymbol{\mu}_i, \tag{13}$$

Applying the Law of Total Variance, the element-wise prior variance for $\mathcal{T}^*$ is:

$$\boldsymbol{\sigma}^{2*} = \sum_{k=1}^K \alpha_k \boldsymbol{\sigma}_k^2 + \sum_{k=1}^K \alpha_k (\boldsymbol{\mu}_k - \boldsymbol{\mu}^*) \odot (\boldsymbol{\mu}_k - \boldsymbol{\mu}^*), \tag{14}$$

where $\odot$ denotes element-wise multiplication. In Eq.14, the first term, which represents the expectation of the conditional variances (computed as a weighted average of the per-task variances), quantifies the uncertainty within each expert task. The second term, which is the variance of the conditional means, quantifies the dispersion or uncertainty among different expert tasks. Eq.13 and Eq.14 together define the task semantic, providing a more informative prior than modeling the parameters as a standard Gaussian, as it ensures that both intra-task variation and inter-task uncertainty are captured when modeling the latent prior for each unseen task.

### A.3 Additional Results on Cross-dataset Image-text Retrieval

As shown in Fig. 6, we present cross-dataset evaluation with the model trained on MS-COCO and tested on Flowers102. It can be observed that when there exists a noticeable gap between the expert task and the inference task, although the performance generally surpasses Zero-Shot CLIP, both AdapterSoup and Top-$K$ LoRA Weighted fail to significantly outperform Model Soup. In contrast, the semantics-guided SG-LoRA demonstrates a strong ability in generating high-performance LoRA parameters.

Table 6: Cross-dataset evaluation with the model trained on MS-COCO and tested on Flowers102. The best results are highlighted in bold, and the second-best results are underlined.

| Method | I2T Metrics | | | T2I Metrics | | |
|---|---|---|---|---|---|---|
| | R@1 | R@5 | R@10 | R@1 | R@5 | R@10 |
| Zero-Shot CLIP | 22.30 | 53.57 | 69.41 | 16.33 | 46.79 | 68.49 |
| Oracle | 26.23 | 59.17 | 77.55 | 21.57 | 57.56 | 80.13 |
| Model Soups | 23.50 | 54.84 | 73.85 | 17.23 | 50.96 | 73.48 |
| AdapterSoup | 24.21 | 54.26 | 72.83 | 17.76 | 50.93 | 73.83 |
| Top-$K$ LoRA Weighted | 23.98 | 52.83 | 71.91 | 17.63 | 50.69 | 73.74 |
| SG-LoRA | **26.83** | **56.63** | **74.16** | **20.52** | **53.71** | **76.69** |

### A.4 Additional Results on Classification Task

In addition to the image-text retrieval task, we also explored the performance of SG-LoRA on classification tasks. Specifically, we selected 20 superclasses from CIFAR-100 Krizhevsky et al. (2009) as 20 distinct tasks, with each task corresponding to a 5-class classification. The superclasses were chosen as defined in the official CIFAR-100 hierarchy. We selected 8 of these tasks as expert LoRAs and performed inference on 6 unseen tasks, with the results presented in Table.7. We observed that, compared to Zero-Shot CLIP, both Model Soups and the selection of expert parameters to construct LoRA improved classification performance on unseen tasks. Furthermore, SG-LoRA achieved the best performance, indicating that our method is also applicable to classification tasks.

However, we found that SG-LoRA still has a significant performance gap compared to fine-tuning LoRA directly on unseen tasks using training data. We hypothesize that this is because, for classification tasks, the independence (or orthogonality, where decision boundaries between tasks may be orthogonal) between tasks is much more pronounced. In contrast, for image-text retrieval tasks, which operate at a finer granularity, there exist stronger inter-task correlations—this inherent property facilitates feasible LoRA parameter transfer. As shown in Figure.8, we visualized the average LoRA parameters for the 20 tasks and found that, although semantically similar tasks are closer in parameter space, their distances remain relatively sparse compared to image-text retrieval tasks in Figure.7. We plan to conduct further exploration on this issue in the future.

Additionally, we conducted an ablation study on textual description for the classification task, as shown in Table.8. It can be observed that, for the current task setting, more detailed textual descriptions, which account for the specific categories within each task, can better capture the semantic relationships between tasks, thus leading to improved performance.

Table 7: Model Performance of image classification on CIFAR-100 superclass.

| Method | Accuracy |
|---|---|
| Zero-Shot CLIP | 72.30 |
| Oracle | 91.43 |
| Model Soups | 75.63 |
| AdapterSoup | 72.60 |
| Top-$K$ LoRA Weighted | 72.70 |
| SG-LoRA | 77.50 |

Table 8: Ablation study of textual description on CIFAR-100 superclass Classification

| Description | Accuracy (%) |
|---|---|
| A photo of a *<superclass name>* | 75.77 |
| This is a classification task for recognizing *<superclass name>*, which includes *class_1, ..., class_5* | 77.50 |

## A.5 MORE ABLATIONS AND ANALYSIS

Table 9: Ablation study on modalities of semantic prior condition

| Condition | Metrics | | Dataset |
|---|---|---|---|
| | I2T R@1 | T2I R@1 | |
| Visual | 73.16 | 52.70 | MS-COCO |
| Textual | 74.31 | 54.42 | |
| Visual | 86.30 | 70.12 | Flickr30K |
| Textual | 86.90 | 70.66 | |

| Condition | Metrics | Dataset |
|---|---|---|
| | Accuracy | |
| Visual | 73.83 | CIFAR-100 |
| Textual | 77.50 | |

**Ablation on modalities of semantic priors.** The construction of semantic priors serves as the foundation for our `SG-LoRA`. In Table.9, we compare the performance of semantic conditions across different modalities, where the conditional task description from each modality directly influences the selection of experts for unseen tasks by affecting $\alpha_k$ in Eq.6. The visual condition is obtained by averaging the visual embeddings of training set images within each task dataset using a frozen CLIP visual encoder. Experimental results show that the textual condition better captures the semantic relationships between tasks. This could be attributed to two factors. Firstly, the high degree of condensation in textual semantics might play a role. Secondly, the disparities between training and test set images (or the presence of noisy images) within a task could result in inaccuracies in the visual prior condition.

**Impact of expert repository configuration.** Consistent with Table 3, we further evaluated the impact of incorporating the *Dog* expert from the MS-COCO dataset on retrieval performance for two unseen dog tasks in the OxfordPets dataset. As demonstrated in Table 10, including the *Dog* expert in the expert repository consistently improves the performance.

**Visulization of LoRA parameters.** To further investigate the parameter diversity of `SG-LoRA`, we conducted evaluations on the unseen '*Zebra*' task from the MS-COCO dataset at different training stages and visualized the generated LoRAs using t-SNE. As shown in Figure 9, we observe that the distribution of LoRAs generated by `SG-LoRA` gradually aligns with that of Oracle LoRAs (directly trained in image-caption pairs), while still preserving diversity rather than extensively overlapping with the Oracle. This indicate that, by injected stochasticity, our method effectively explores the high-performance LoRAs in the parameter space.

Additionally, by examining the bottom subfigure, we observe that in parameter space, both Adapter-Soup (Tok-$K$ LoRA Merging) and Tok-$K$ LoRA Weighted lie closer to the mean of the Oracle LoRA compared to Model Soup. This is because the latter treats all experts equally, whereas the former two provide more informative semantic guidance, allowing the LoRA parameters to be better tailored to the current unseen task.

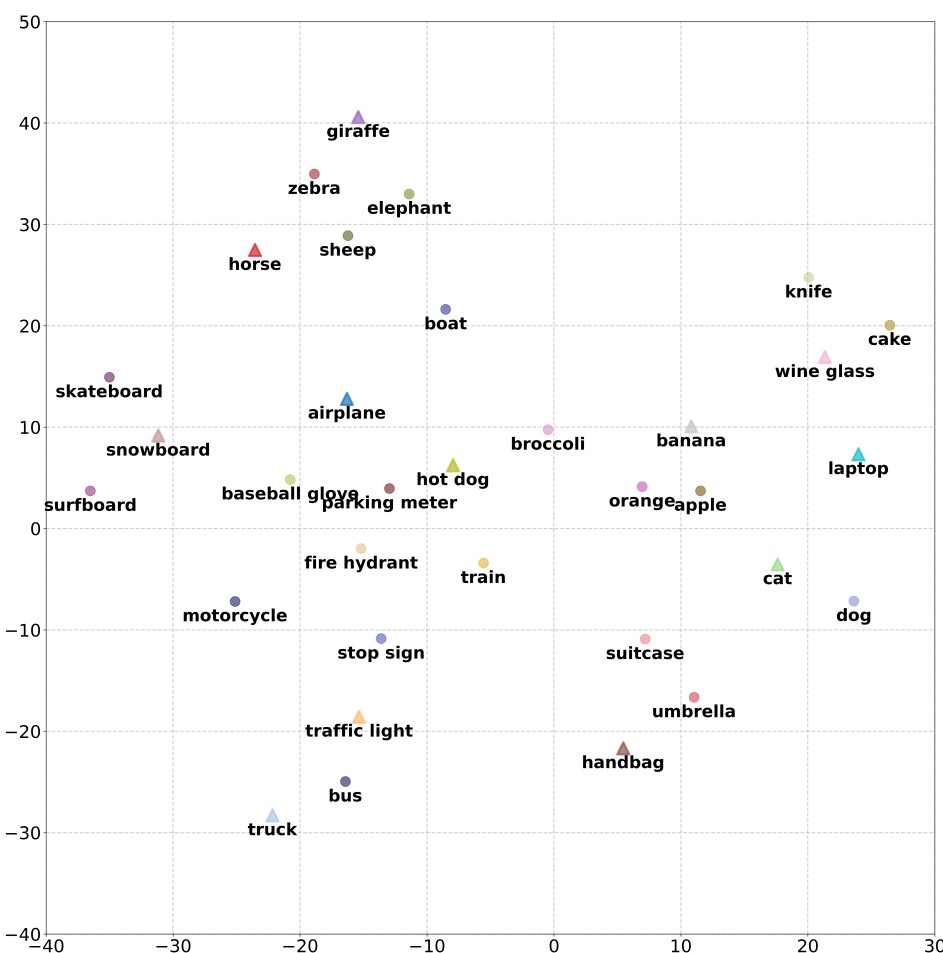

Figure 7: t-SNE visualization of the averaged LoRA parameters on MS-COCO dataset for image-text retrieval task. Triangular markers indicate expert LoRAs. Semantically similar LoRA parameters tend to cluster closely together.

Table 10: Ablation study on expert repository strategy for cross-dataset evaluation. We assess the impact of the *Dog* expert from MS-COCO dataset on the retrieval performance in two unseen dog tasks from OxfordPets dataset (marked with gray text).

| Expert strategy | I2T Metrics | | | T2I Metrics | | |
|---|---|---|---|---|---|---|
| | R@1 | R@5 | R@10 | R@1 | R@5 | R@10 |
| *Pug* | | | | | | |
| w/o *Dog* expert | 45.00 | 66.00 | 78.00 | 32.00 | 51.00 | 61.25 |
| w *Dog* expert | 46.00 | 67.00 | 78.00 | 32.00 | 51.75 | 62.75 |
| *Chihuahua* | | | | | | |
| w/o *Dog* expert | 64.00 | 89.00 | 93.00 | 52.50 | 81.00 | 88.25 |
| w *Dog* expert | 66.00 | 90.00 | 95.00 | 55.25 | 81.25 | 89.75 |

## A.6 LIMITATIONS AND FUTURE WORK

Although our method achieves strong performance, there remain several directions for future exploration. First, the current expert repository assumes that all LoRA experts share a same structure, such as having the same rank. In practice, however, publicly available expert LoRAs may be heterogeneous.

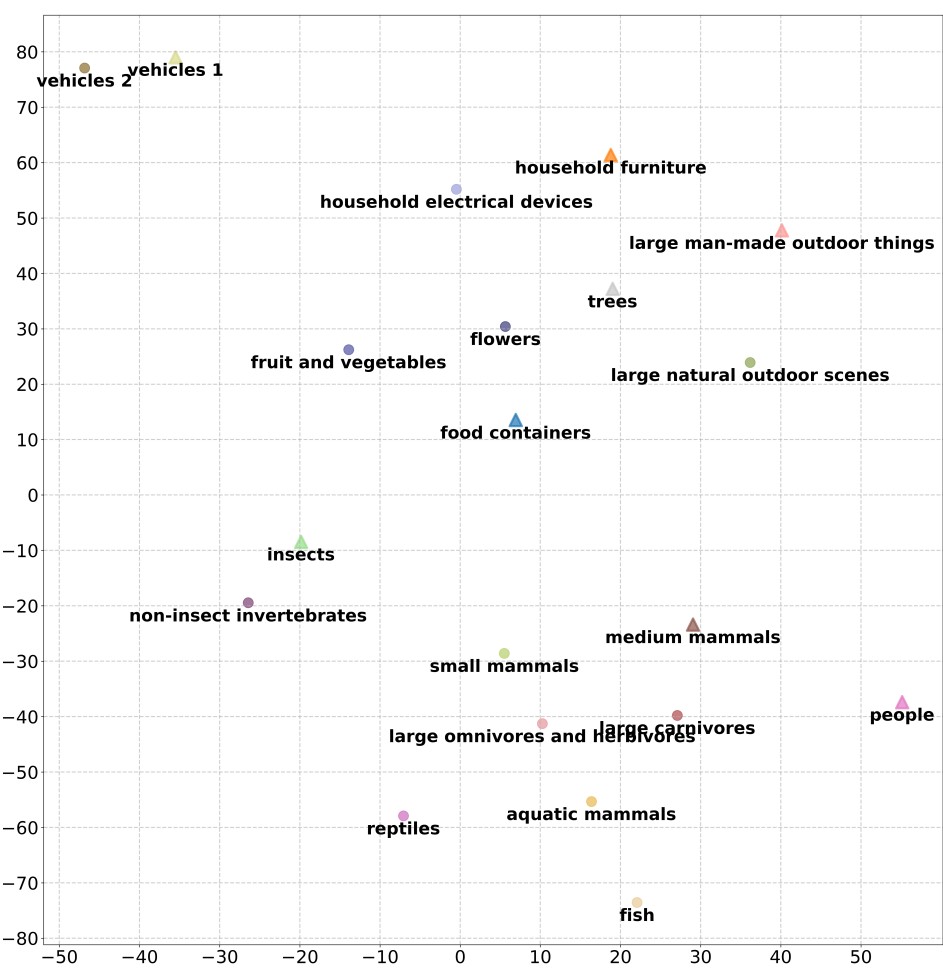

Figure 8: t-SNE visualization of the averaged LoRA parameters on CIFAR-100 for classification task. Triangular markers indicate expert LoRAs. Although semantically similar LoRA parameters tend to cluster closely together, the distribution appears sparser compared to the MS-COCO dataset due to stronger task independence.

Extending the conditional LoRA generation process to account for such heterogeneity could enable the integration of a broader range of expert knowledge. Moreover, our current approach employs the standard CLIP textual template as the task description. Leveraging task descriptions derived from LLM-based reasoning may provide more accurate and semantically aligned guidance.

### A.7 THE USE OF LARGE LANGUAGE MODELS

In this work, Large Language Models (LLMs) were used exclusively for language polishing and spelling correction.

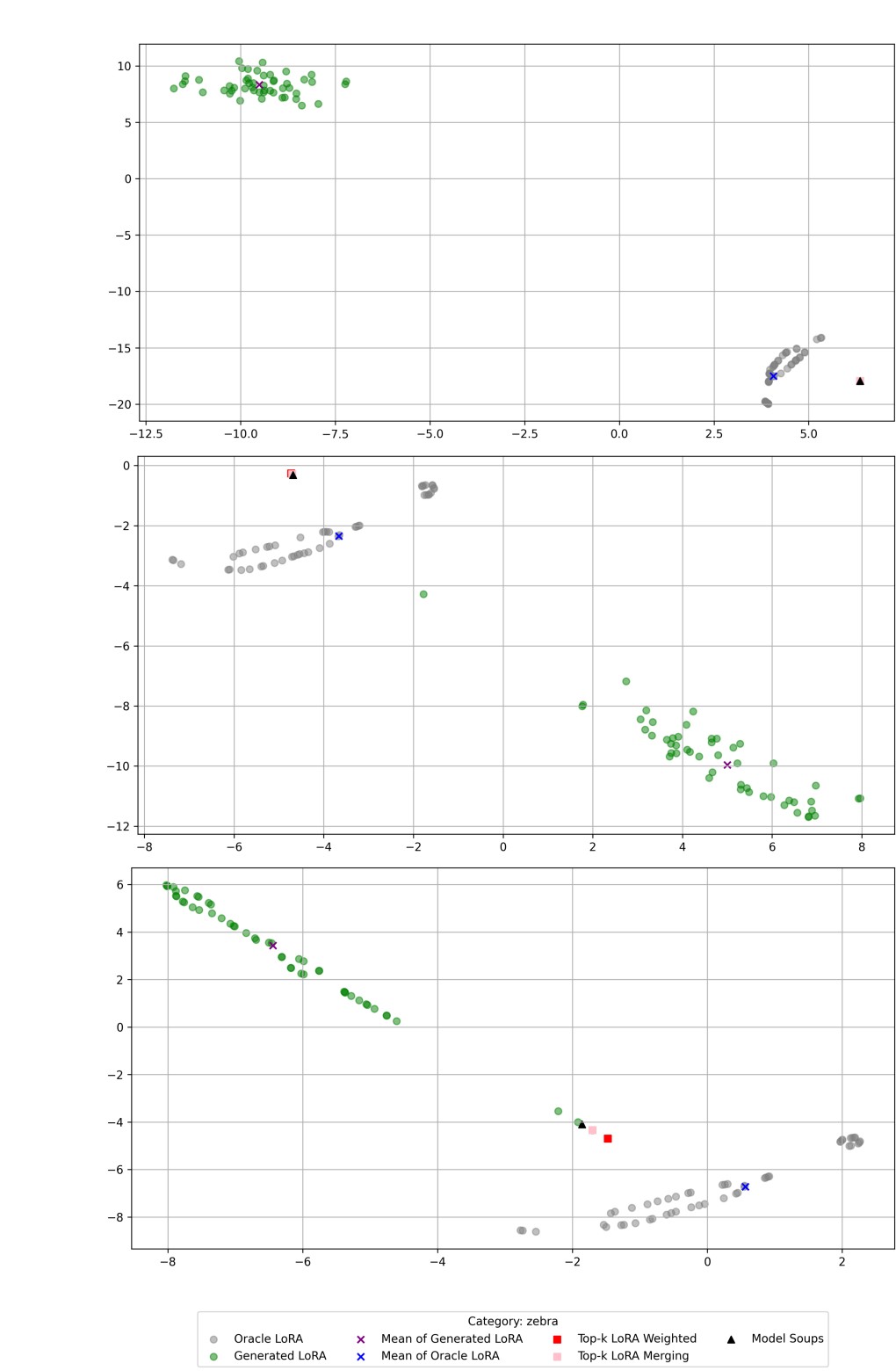

Figure 9: t-SNE visualization of LoRA parameters using different comparative methods, tested on the unseen 'Zebra' category at different training stages. For SG-LoRA and Oracle LoRA, we randomly sampled 50 samples each. The subfigures from top to bottom represent increasing CVAE training epochs.

