# OpenReview forum: "SEMANTIC-GUIDED LORA PARAMETERS GENERATION"
_ICLR.cc/2026/Conference — ICLR 2026 Conference Withdrawn Submission_

### Official Review · Reviewer_4TZX · 2025-10-20

**Soundness:** 3
**Presentation:** 3
**Contribution:** 3
**Rating:** 6
**Confidence:** 3

**Summary:**

The paper introduces Semantic-Guided LoRA Parameter Generation (SG-LoRA), a tuning-free generative framework that produces task-specific LoRA parameters for unseen tasks using only semantic task descriptions. It defines a new setting called Zero-Shot Open-World Adaptation (ZSOA), where models must adapt to entirely new tasks without data access. SG-LoRA leverages task semantics—encoded through CLIP text embeddings—and a conditional variational autoencoder (CVAE) to generate LoRA parameters conditioned on semantically relevant expert LoRAs. The approach demonstrates strong performance across image-text retrieval benchmarks and extends to classification tasks.

**Strengths:**

1) The paper tackles an important problem by introducing a scalable generative framework for LoRA parameter synthesis without any tuning or data access.

2) The proposed semantic-guided CVAE formulation is technically sound and effectively models task-specific priors, enhancing both parameter diversity and robustness to domain shifts.

3) The experiments demonstrates consistent improvements over baselines and even outperforming oracle fine-tuning in some cases, which highlights the method’s strong generalization ability.

**Weaknesses:**

1) Even though SG-LoRA eliminates the need for direct data access from expert tasks, it still assumes the availability of accurate and descriptive task text annotations to construct semantic priors. In realistic deployment scenarios, many publicly released LoRA modules are shared without detailed metadata or task descriptions, often lacking information about their training domains or objectives. This raises concerns about the practicality of applying SG-LoRA when the semantic linkage between tasks cannot be explicitly derived. The paper would benefit from discussing how the framework could operate or be extended when such semantic descriptions are missing or noisy.

2) The paper does not provide comparisons against more parameter-generation baselines, specifically the approaches mentioned in Section 2.3, making it difficult to gauge the full novelty margin.

**Questions:**

Please address concerns in Weaknesses above.

---

### Official Review · Reviewer_GGJA · 2025-10-24

**Soundness:** 3
**Presentation:** 2
**Contribution:** 3
**Rating:** 4
**Confidence:** 4

**Summary:**

This paper proposes SG-LoRA, a framework for semantic-guided lora parameter generation. This paper introduces a new task setting:zero-shot open-world (ZSOA) adaptation setting. The major idea is to pre-train multiple lora experts on source tasks, represent each task via semantic embeddings, aggregates top-K relevant experts to form a prior gaussian distribution, and employs a conditional VAE to generate new lora weights for unseen tasks based on their task descriptions. Experiments on multiple cross-modal retrieval tasks show that SG-LoRA surpasses fusion-based baselines.

Overall, the paper presents several inspiring ideas and a promising framework for zero-data adaptation, but its writing, notation, and experimental design require substantial refinement. If the authors can address these issues, I would be inclined to raise my score.

**Strengths:**

- The proposed zero-shot open-world adaption is a practical and meaningful task setting, which extends conventional domain adaption toward semantic-driven parameter generation.
- The overall architecture, which integrates semantic aggregation and generative lora modeling is intuitive and insightful.
- Generating parameters for unseen tasks without target data has significant potentials in real-world applications.
- The proposed method outperforms older static-fusion baselines.

**Weaknesses:**

- The conceptual framing of Zero-Shot Open-World Adaptation (ZSOA) appears overstated. The experiments are still standard text–image retrieval on new datasets, which aligns more closely with zero-shot domain or data transfer rather than true task-level adaptation.
- The decision to employ a CVAE for parameter generation is insufficiently justified, both theoretically and empirically. It remains unclear why a probabilistic generative model is necessary compared to simpler deterministic model.
- Some methodological choices are not clearly explained. In particular, the use of epoch-averaged LoRA parameters (Eq. 3 & 7) instead of the final checkpoint is not clearly motivated or empirically validated.
- The weighting coefficient λ in the ELBO objective is introduced without analysis or sensitivity evaluation, leaving its influence on model performance uncertain.
- Several variables (e.g., ΔW, X, μ, c) are introduced informally and used inconsistently across sections, which makes the mathematical exposition hard to follow.
- The paper mainly compares against older fusion-based approaches such as ModelSoup and AdapterSoup (2022–2023), while omitting more recent generative or meta-learning-based LoRA methods. As a result, the advantage of SG-LoRA over the current state-of-the-art remains unclear.

**Questions:**

- Could the authors elaborate on what specifically distinguishes ZSOA from the standard zero-shot domain or dataset transfer and provide more evidence to justify the "task-level generalization" claim.
- What is the motivation for employing a conditional VAE as the parameter generator? What is the advantage? Have the authors considered or compared other generative models (e.g., diffusion model) or simpler deterministic alternatives?
- Equations (3) and (7) indicate that LoRA parameters are averaged across M epochs. Could the authors clarify why this averaging is preferable to using the final checkpoint?
- How sensitive is the model performance to the weighting coefficient $\lambda$ in the ELBO objective? Was $\lambda$ tuned or fixed, and how does it influence the trade-off between reconstruction and regularization?
- It would strengthen the paper if a comparison against current SOTA methods is provided.

---

### Official Review · Reviewer_Vco4 · 2025-11-01

**Soundness:** 2
**Presentation:** 3
**Contribution:** 2
**Rating:** 2
**Confidence:** 3

**Summary:**

SG–LoRA addresses zero-shot/open-world construction of LoRA weights for unseen tasks by learning a generative mapping from semantic task descriptions (task-level text or intent embeddings) to LoRA parameter distributions. The pipeline uses a shared semantic embedding space to measure proximity to known expert tasks and distills knowledge to generate LoRA modules for new tasks without task-specific data (a “no raw data” zero-shot setting). Results are reported on multiple tasks and claim strong adaptability and privacy-preserving benefits.

**Strengths:**

Tackles a timely deployment problem: how to synthesize adapters for unseen user intents without exposing raw data. This has privacy and efficiency implications.

The semantic-to-parameter pipeline is an elegant framing: mapping from task-description embeddings to parameter space is a compelling approach for zero-shot adapter synthesis.

**Weaknesses:**

Generative model stability and calibration: the paper does not convincingly show how it avoids producing harmful or nonsensical weights for out-of-distribution semantics. No confidence estimation or rejection mechanism is described.

Evaluation scope: quantitative evaluation relies on a curated set of “expert tasks” - it is unclear how well the generator generalizes to truly novel or adversarial intents. More OOD testing is needed.

Potential ethics/risks: automatically generating adapters that alter model behavior might enable stealthy model misuse (e.g., creating adapters that bias outputs). Ethical discussion and safeguards are thin.

**Questions:**

How is semantic proximity measured and how sensitive are generated parameters to variations in the text description? Provide calibration plots.

What mechanisms prevent the generator from producing destructive or biased adapters when given malicious or out-of-distribution prompts? Are there reject options or uncertainty thresholds?

Can the authors provide more ablations on the generator architecture (autoregressive vs. conditional normalizing flow vs. deterministic mapping) and on training objectives (KL, MSE, adversarial losses)?

---

### Official Review · Reviewer_fwaD · 2025-11-01

**Soundness:** 3
**Presentation:** 3
**Contribution:** 3
**Rating:** 4
**Confidence:** 3

**Summary:**

This paper introduces Semantic-guided LoRA Parameter Generation (SG-LoRA), a novel framework for Zero-Shot Open-world Adaptation (ZSOA). ZSOA is a challenging setting where a model must generate task-specific Low-Rank Adaptation (LoRA) parameters for entirely unseen tasks, without access to any task data and within an unconstrained task space. The core idea is to use task descriptions as a semantic bridge. By measuring the semantic proximity between a new task's description and those of a set of pre-trained "expert" LoRAs in a shared CLIP embedding space, SG-LoRA constructs a task-specific prior. This prior then guides a Conditional Variational Autoencoder (CVAE) to generate the LoRA parameters for the novel task directly, enabling real-time, tuning-free personalization of large pre-trained models.

The primary contributions are threefold. First, the authors formalize the ZSOA problem, a more realistic and demanding scenario than typical zero-shot learning. Second, they propose the SG-LoRA framework, which uses a sparse aggregator to select the most relevant expert LoRAs and a CVAE to model the parameter distribution for generation, introducing beneficial stochasticity. Finally, extensive experiments on image-text retrieval and classification benchmarks demonstrate that SG-LoRA can generate high-performance LoRA parameters that are competitive with, and sometimes even surpass, those obtained by traditional fine-tuning (the Oracle method), while offering advantages in privacy, scalability, and flexibility for edge deployment.

**Strengths:**

1. Novel Problem Formulation and Practical Relevance: The paper introduces a highly relevant and challenging task, Zero-Shot Open-world Adaptation (ZSOA), which closely mirrors real-world deployment needs, especially on resource-constrained edges. This setting, which requires generalization to unseen tasks without any data in an unconstrained task space, is more realistic and demanding than conventional zero-shot learning or model merging.

2. Elegant and Effective Framework Design: The core idea of SG-LoRA is clever and well-executed. Using task descriptions as a semantic bridge to guide knowledge transfer from expert LoRAs to novel tasks is a powerful approach. The two-stage design—employing a sparse aggregator for relevant expert selection and a Conditional VAE for stochastic parameter generation—enables tuning-free inference and introduces beneficial diversity, enhancing robustness and adaptability.

3. Comprehensive and Convincing Empirical Validation: The paper provides extensive experiments across multiple benchmarks (in-domain, cross-dataset retrieval, and classification). The results robustly demonstrate that SG-LoRA can generate high-performance LoRA parameters that are competitive with, and sometimes surpass, those from traditional fine-tuning. The ablations and analysis effectively validate the contribution of each component.

**Weaknesses:**

1. Assumption of Homogeneous LoRA Structure: A significant limitation is the assumption that all expert LoRAs share an identical structure (e.g., the same rank r). In practice, the ecosystem of public LoRAs is highly heterogeneous. The current framework cannot integrate these structurally diverse experts, which limits its applicability in a truly open-world setting.

2. Notable Performance Gap on Classification Tasks: While the method excels in image-text retrieval, its performance on the CIFAR-100 classification benchmark shows a significant gap compared to the Oracle (direct fine-tuning). This suggests the approach may be most effective for tasks with strong inter-task correlations (like retrieval) and less so for tasks with more independent or orthogonal decision boundaries, revealing a boundary to its generalization.

3. Dependence on Description Quality and Expert Curation: The performance is inherently dependent on two external factors: the quality of the task description (relying on a simple template may be insufficient for complex tasks) and the composition and coverage of the expert repository. Performance may degrade if the repository does not adequately cover the semantic neighborhood of a new task. While the paper shows flexibility with mixed repositories, the optimal construction and maintenance of this expert base remain an open challenge.

**Questions:**

1. Zero-shot CLIP fails to leverage the numerous pre-trained LoRAs; it would be better to focus on more targeted methods that integrate LoRAs.
2. It is possible to test on more open out-of-sample data. Modules trained on a few centralized datasets may not be able to reflect the performance of zero-shot datasets.

---

### Note · Authors · 2025-11-14

I have read and agree with the venue's withdrawal policy on behalf of myself and my co-authors.